# The Effect of Sn Addition on Zn-Al-Mg Alloy; Part I: Microstructure and Phase Composition

**DOI:** 10.3390/ma14185404

**Published:** 2021-09-18

**Authors:** Peter Gogola, Zuzana Gabalcová, Martin Kusý, Henrich Suchánek

**Affiliations:** Institute of Materials Science, Faculty of Materials Science and Technology in Trnava, Slovak University of Technology in Bratislava, Ulica Jána Bottu 25, 917 24 Trnava, Slovakia; zuzana.gabalcova@stuba.sk (Z.G.); martin.kusy@stuba.sk (M.K.); henrich.suchanek@stuba.sk (H.S.)

**Keywords:** Zn-based alloy, phase composition, XRD, DSC, microstructure formation, Sn-addition, intermetallic phases

## Abstract

In this study, the addition of Sn on the microstructure of Zn 1.6 wt.% Al 1.6 wt.% Mg alloy was studied. Currently, the addition of Sn into Zn-Al-Mg based systems has not been investigated in detail. Both as-cast and annealed states were investigated. Phase transformation temperatures and phase composition was investigated via DSC, SEM and XRD techniques. The main phases identified in the studied alloys were η(Zn) and α(Al) solid solutions as well as Mg_2_Zn_11_, MgZn_2_ and Mg_2_Sn intermetallic phases. Addition of Sn enabled the formation of Mg_2_Sn phase at the expense of Mg_x_Zn_y_ phases, while the overall volume content of intermetallic phases is decreasing. Annealing did not change the phase composition in a significant way, but higher Sn content allowed more effective spheroidization and agglomeration of individual phase particles.

## 1. Introduction

Zn-Al alloys are used as corrosion protection coatings for a series of applications including steel strands used to reinforce overhead power lines. Such strands ensure the overall mechanical rigidity of aluminum conductors and thus reduce the number of supporting towers needed for a specific distance of overhead power lines [1]. These power lines are designed to operate at about 180 °C, while due to an increased current load, they may heat up to 300 °C. [1,2] Pure Zn coatings are not suitable in such conditions. Above 200 °C, the pure Zn coating starts to react with the steel [1,3,4] substrate and continues to form ZnFe intermetallics. Such a reaction can reduce the actual steels cross section area and thus reduce the cables’ mechanical properties. This limitation can be overcome by alloying [5,6].

Nowadays, various Zn-Al-Mg alloy system coatings are available, such as the commercially well-known Magizinc (MZ) with a chemical composition of Zn 1.6 wt.% Al 1.6 wt.% Mg [7,8,9,10,11,12,13,14].

Mg is added to Zn-based coatings to further increase their corrosion protection capabilities. Mg is useful especially for increasing the galvanic protection offered by a coating at cut-edges and mechanically damaged spots. Additional alloying may further improve the properties of Zn-Al-Mg alloys as reported by several literature sources [15,16,17,18,19,20,21,22,23,24,25]. Unfortunately, there is still lack of information about Sn addition into these systems [26].

In recent years, the effects of Sn addition on the microstructure of Mg- and Al-based alloys has been studied. This includes microstructural stability upon thermal exposure. Sn is known to have a high affinity to Mg, creating mainly the Mg_2_Sn intermetallic phase. All in all, a positive influence of Sn on Mg and Al-based alloys has been reported depending on the amount of Sn added to these alloys. Alloying enabled the formation of Mg_2_Sn in all these alloys [27,28,29,30,31,32,33]. The aim of this research is to confirm if Mg_2_Sn phase is also preferred compared to Mg_x_Zn_y_ phases in the current alloy system as suggested by studied literature sources [32,34]. Phase composition and overall microstructure character will be investigated on as-cast samples.

As one of the potential applications involves a long-term thermal exposure, microstructure and phase composition will be investigated after a representative annealing treatment of each alloy as well. Experimental annealing temperature will be set to 310 °C to be clearly above potential exposure temperatures, but below the melting temperatures (~340 °C) of the investigated alloys.

Mg_2_Zn_11_ and MgZn_2_ are the most common intermetallic phases in the currently investigated Zn-based systems. MgZn_2_ is reported to be less ductile, while both are reported to be slightly less ductile compared to Mg_2_Sn [18,19,20,29,35,36]. The possibility to replace Mg_x_Zn_y_ phases by Mg_2_Sn and thus reduce the overall volume content of intermetallic phases in a Zn 1.6 wt.% Al 1.6 wt.% Mg based alloy would indicate an interesting research path for further extensive investigation of corrosion and mechanical properties of such alloys.

## 2. Materials and Methods

Five different alloys with the designed nominal composition of Zn-1.6Al-1.6Mg-xSn (wt.%), where x = 0.0, 0.5, 1.0, 2.0 and 3.0 wt.%, respectively, were prepared by melting pure Zn at 470 °C and mixing in the appropriate amount of a 50 wt.% Al and 50 wt.% Mg master alloy. These raw materials were preheated to 400 °C to facilitate their rapid melting. Due to the low melting point of Sn, it was added in the last step. Table 1 indicates that the measured bulk chemical compositions of the alloys by glow discharge optical emission spectroscopy (GDOES, Spectruma GDA 750, Spectruma Analytik GmbH, Hof, Germany) are in good agreement with the nominal ones.

Casting was done from 470 °C of melt temperature into a water-cooled copper mold with a diameter of 30 mm and depth of 20 mm. During casting, the sample temperature was continuously measured at a sampling frequency of 25 Hz with K-type thermocouples attached to the mold surface. The cooling rate of 60 °C/s was established.

Two types of cylindrical samples were prepared for each alloy: (i) as-cast samples and (ii) cast and subsequently solution annealed at 310 °C for 1 h. The annealing step was finished by ice-water quenching. A cooling rate of 75 °C/s was recorded. The selected solution annealing temperature corresponds to the γ + η region of the Zn-Al system [37], while it is clearly below the melting point of all chosen alloys. Cooling in cold water ensured a very good control of the annealing time.

Vickers hardness tests were carried out in line with ISO 6507-1 [38] on polished surfaces of as-cast and annealed samples via a BUEHLER Indentamet 1105 (Buehler Ltd., Lake Bluff, IL, USA) hardness tester at an applied load of 9.8 N, holding time at the point of load application was 10 s.

DSC measurements were carried out by the Perkin Elmer Diamond DSC (Perkin Elmer Inc., Billerica, MA, USA) device. The DSC samples were cut from as-cast samples to a target weight of 5 mg. The samples were heated to the temperature of 500 °C at a heating rate of 10 °C/min and then cooled to ambient temperature at a cooling rate of 10 °C/min under the protective argon atmosphere.

The XRD analysis was carried out on metallic filings of the as-cast and annealed samples by the PANalytical Empyrean X-ray diffractometer (XRD) (Malvern Panalytical Ltd., Malvern, UK). The procedure to measure on metallic filings instead of bulk castings was chosen to limit the influence of casting texture on the recorded XRD pattern. The casting texture added additional complexity to the XRD measurements by influencing the theoretical relative intensities for the individual crystallographic planes. This made the quantitative analysis very unreliable due to the complex texture corrections needed. The measurements were performed in Bragg–Brentano geometry. Theta-2Theta angle range between 10° and 148° 2Theta was chosen. The XRD source was set to 40 kV and 40 mA. The incident beam was modified by 0.04 rad soller slit, 1/4° divergence slit and 1/2° anti-scatter slit. The diffracted beam path was equipped with a 1/2° anti-scatter slit, 0.04 rad soller slit, Ni beta filter and PIXcel3D position sensitive detector operated in 1D scanning mode. The phase quality was analyzed using PANalytical Xpert High Score program (HighScore Plus version 3.0.5) with the ICSD FIZ Karlsruhe database. Quantitative results were determined from XRD patterns using the Rietveld refinement-based program MAUD version 2.84 [39]. The program uses an asymmetric pseudo-Voight function to describe experimental peaks. Instrument broadening was determined by measuring the NIST660c LaB_6_ (The National Institute of Standards and Technology, Gaithersburg, MD, USA) line position and line broadening standard and introduced to the Rietveld refinement program via the Caglioti equation. Anisotropy size-strain model was applied to Zn solid solution while other phases were treated by isotropic models. A minor discrepancy between the nominal and measured peak intensities was corrected using the spherical harmonic functions with fibre symmetry. The quality of the fit was in all analyzed samples below 10% R_wp_.

The metallographic preparation of DSC, as-cast and annealed samples consisted of standard grinding using abrasive papers and polishing on diamond pastes with various grain sizes of down to 0.25 μm.

The microstructure evaluation was performed by the JEOL JSM 7600F scanning electron microscopy (SEM, Jeol Ltd., Tokyo, Japan) with a Schottky field emission electron source operating at 20 kV and 90 µA. The samples were placed at a working distance of 15 mm and documented using a backscattered electron detector. The chemical element analysis was performed via the Oxford Instruments X-Max silicon drift detector, energy dispersive X-ray spectrometer (EDS, Oxford Instruments plc, Abingdon, UK).

Image analysis was performed on at least 15 sites for each sample by ImageJ FIJI 1.53c software [40]. Area ratio of η(Zn) based areas and other microstructure components were established.

All results are listed as the average values of multiple measurements with ± standard deviation error bars.

## 3. Results

The DSC curves of MZ + xSn alloys system in Figure 1a–e show charts of DSC heating runs. Figure 1 shows the most relevant section of the recorded data, while measurements were done from 20 to 500 °C at a heating rate of 10 °C/min. The first heating runs recorded are presented to observe the reactions in the as-cast samples during heating and subsequent melting, including eutectoid reactions (not visible in DSC cooling runs).

Hence, the individual microstructure features in the as-cast samples cannot be clearly distinguished; the DSC samples after the cooling run were investigated via SEM. These observations enabled us to identify the individual reactions for each recorded peak.

Peaks were recorded describe the reactions of the following phases: η(Zn)–hcp Zn-based solid solution; γ(Al)–fcc Al-based solid solution present above the eutectoid reaction temperature reported at 275 °C in the Zn-Al system [37] up to melting temperature; α(Al)–fcc Al-based solid solution present below the Zn-Al system eutectoid reaction; Mg_2_Zn_11_, MgZn_2_ and Mg_2_Sn intermetallic phases of respective systems.

The curve in Figure 1a corresponds to the MZ + 0.0Sn alloy. The first peak observed at about 285.0 °C corresponds to the eutectoid transformation α(Al) + η(Zn) → γ(Al). This peak is repeated for all alloys observed. Melting of pure MZ + 0.0Sn starts at 344.0 °C (peak maximum at 347.5 °C) with melting of the ternary eutectic consisting of η(Zn), γ(Al) and Mg_2_Zn_11_ phases. An example of such areas is provided in Figure 2a. This is followed by the melting of the Zn/Mg_x_Zn_y_ binary eutectic (peak maximum at 357.0 °C). A clear example can be observed in Figure 2b. Mg_x_Zn_y_ corresponds to a mixture of Mg_2_Zn_11_ and MgZn_2_ phases as documented in Figure 2c. A closer detail of this area is given in Figure 2d) showing also the α(Al) + η(Zn) eutectoid particles in detail. Chemical composition of the individual phases is documented in Table 2. The last peak corresponds with the melting of the Zn rich dendrites (peak maximum at 378.9 °C) [41].

As indicated in Figure 2a, η(Zn) dendrites are always decorated by a needle like α(Al) particles. These are formed as a result of the decreasing solubility of Al in Zn in the temperature range below ~285 °C (see Figure 1). These particles are observed for all alloys investigated.

As observed in Figure 1b, the addition of 0.5 wt.% of Sn enables the formation of a peak at 335 °C. This effect corresponds to newly emerging quaternary eutectic areas consisting of η(Zn), γ(Al), Mg_2_Zn_11_ and Mg_2_Sn phases. A typical such area is shown in Figure 3a) with a selected detail in Figure 3b. The chemical composition of the Mg_2_Sn phase was measured and listed in Table 2. These areas start to melt at about 334 °C (peak maximum at 335.0 °C). The next peak corresponds to the melting of the ternary eutectic (342.0 °C) composed of η(Zn), γ(Al) and Mg_2_Zn_11_. The adjacent peak indicates the melting of the η(Zn) + Mg_2_Zn_11_ binary eutectic (353.9 °C). The peak at 378.3 °C indicates the melting of the remaining η(Zn) dendrites.

The alloy with 1 wt.% of Sn (Figure 1c) has a minor peak left corresponding to the ternary eutectic (339.1 °C), while the peak corresponding to the quaternary eutectic (335.3 °C) increased further in peak area. Both other peaks correspond to the same reactions as described above.

With 2 wt.% of Sn (Figure 1d), the melting starts at 333.5 °C. This reaction melts the complex eutectic area shown in Figure 4. The peak corresponding to the ternary eutectic reaction is not resolved separately anymore. The peak at 349.6 °C in this alloy represents the melting of two binary eutectics: η(Zn) + Mg_2_Zn_11_ as well as η(Zn) + Mg_2_Sn. The last peak in this DSC curve at 373.4 °C corresponds again to η(Zn) dendrites.

Adding 3 wt.% of Sn (Figure 1e) changes the peak of the quaternary eutectic reaction only slightly (peak maximum at 336.8 °C). The peak observed at 346.3 °C corresponds according to microstructure observations solely to the melting of the binary η(Zn) + Mg_2_Sn eutectic. Melting of the remaining (η)Zn is indicated by the peak at 372.6 °C.

XRD measurements were performed on the metallic filings prepared from the bulk samples. Figure 5 and Figure 6 shows the XRD patterns for selected alloys in the as-cast and annealed state, respectively. A quantitative analysis using the Rietveld method was performed considering the phases listed in Table 3 characterized in the ICSD FIZ Karlsruhe database. These phases enabled the identification of all significant peaks in the measured XRD patterns.

As the cooling speed in all experiments was rather high at 60–75 °C/s for both as-cast and annealed samples, it was assumed that the solubility changes below the eutectoid transformation [γ(Al) → αAl + η(Zn) at 275 °C] will be significantly limited. Such a phenomenon was reported for α(Al) as well as η(Zn) phases by Gogola et al. [42] based on XRD measurements of Zn-Al based samples. To enable the correct quantitative analysis of α(Al) as well as η(Zn) phases, their chemical composition had to be changed by adding 14 at.% of Zn and 2 at.% of Al, respectively, as suggested in this publication. The chemical composition of the phases was changed in MAUD software before the quantitative analysis of each XRD pattern. The soundness of this approach was double checked comparing the GDOES chemical composition data with the chemical composition calculated from XRD quantitative analysis for each sample.

The volume content of individual phases in the as-cast samples evolved as shown in Figure 7.

In all alloys, both Mg_2_Zn_11_ as well as the non-equilibrium MgZn_2_, the intermetallic phases can be detected. The overall content of Mg_x_Zn_y_ intermetallic phases was reduced from 23.5 vol.% to about 1.5 vol.% by adding 3 wt.% of Sn into the basic Zn 1.6 wt.% Al 1.6 wt.% Mg (MZ) alloy. At the same time, the Mg_2_Sn phase occupied about 8 vol.% of the as-cast MZ + 3.0Sn alloy.

The content of MgZn_2_ was reduced from 3.5 vol.% to ~1.5 vol.% by adding 0.5 wt.% of Sn. The further addition of Sn did not change the content of this phase significantly. Its content was gradually further reduced to ~1 vol.% by adding up to 3 wt.% of Sn into the alloy. However, at ~1 vol.% of MgZn_2_, the detectability limit of MgZn_2_ was likely reached in the current alloy with the applied measurement setup.

Mg_2_Zn_11_ phase was detected in all alloys. Its content was gradually reduced from ~20 vol.% down to below 1 vol.% by adding up to 3 wt.% of Sn.

Peaks corresponding to Mg_2_Sn can be already clearly identified in the as-cast MZ + 0.5Sn sample representing as low as 1.5 vol.% of this phase. Its volume content clearly gradually increased up to ~8 vol.% when 3 wt.% of Sn was added.

The content of α(Al) is calculated to be 2.5 vol.% on average across all alloys investigated. Addition of Sn did not change the content of α(Al) in a significant way.

Annealing the investigated alloys clearly influenced their phase composition (Figure 8) as calculated from XRD measurements (Figure 6).

In the MZ + 0.0Sn alloy, the content of non-equilibrium MgZn_2_ is significantly reduced after annealing. Its content is reduced from ~3.5 vol.% to below 1 vol.%. For all the other alloys, the content of MgZn_2_ is rather similar in both as-cast and annealed states.

The content of the Mg_2_Zn_11_ phase increases to ~28 vol.% after annealing the MZ + 0.0Sn alloy, hence indicating that this is the equilibrium phase for this alloy [34,37]. Additionally, for all the other compositions, the volume content of Mg_2_Zn_11_ increases after annealing. Annealing changes the content of Mg_2_Sn only slightly. Most noticeably, its content reduces at 2 and 3 wt.% of Sn, probably in favor of Mg_2_Zn_11_. The content of α(Al) remains basically unchanged by the annealing process.

Comparison of microstructure images for the most important edge cases is given in Figure 9a–f. Figure 9a,c,e correspond to as-cast states, while Figure 9b,d,f correspond to the annealed state.

The microstructure of MZ + 0.0Sn samples is formed by η(Zn) phase dendrites, where the interdendritic areas are formed by a mixture of binary η(Zn)/Mg_x_Zn_y_ eutectic and ternary η(Zn)/α(Al)/Mg_x_Zn_y_ eutectic [43,44]. Adding 0.5 wt.% of Sn changes the microstructure appearance in an insignificant way (Figure 9a vs. Figure 9c). On the other hand, in Figure 9e, we can clearly observe the presence of η(Zn)/Mg_2_Sn binary eutectic regions. Gradual addition of Sn reduces the amount of (η)Zn/Mg_x_Zn_y_ eutectic regions and gives rise to η(Zn)/Mg_2_Sn eutectic regions. Additionally, areas formed by η(Zn)/α(Al)/Mg_x_Zn_y_ ternary eutectic are reduced in favor of probably η(Zn)/α(Al)/Mg_x_Zn_y_/Mg_2_Sn quaternary eutectics.

Figure 9b,d,f shows the microstructure of selected alloys after annealing. η(Zn) loses its dendritic character as well as all interdendritic areas being spheroidized, while areas with common chemical compositions are connected. None of the previously described eutectic regions can be recognized. Based on XRD, the vast majority of Mg_x_Zn_y_ particles are corresponding to Mg_2_Zn_11_. As annealing was done above the eutectoid temperature of the Zn-Al system (285 °C as reported by DSC measurements, Figure 1), Al rich particles were spheroidized as γ(Al) particles. Hence, the outer shape of Al rich particles remained frozen while decomposition to α(Al) + η(Zn) eutectoid particles took place upon cooling from annealing temperature.

Figure 10 summarizes the vol.% of all other microstructure components apart from η(Zn).

For as-cast samples, this represents the interdendritic spaces which are formed mainly by various eutectics including a certain portion of η(Zn) phase solidified within them as well as α(Al) + η(Zn) eutectoid particles.

For the annealed samples, it was possible to clearly distinguish between η(Zn) matrix and all intermetallic phase particles along with α(Al) + η(Zn) eutectoid particles.

The difference between data for as-cast and annealed samples is mainly caused by the fact that η(Zn) solidified in the interdendritic spaces of the as-cast samples cannot be separately identified, while during annealing, these small η(Zn) particles are allowed to connect to the larger primary η(Zn) areas forming a uniform matrix.

In general, the vol.% of all other microstructure components outside of η(Zn) is decreasing with the addition of Sn into the investigated alloys.

Microhardness was measured on the as-cast and the corresponding annealed samples as well. All samples showed an over twice higher hardness compared to pure Zn. MZ + 0.0Sn showed the hardness of ~113 HV, while the gradual addition of Sn was almost linearly decreasing the alloy’s hardness down to ~85 HV measured on the MZ + 3.0Sn as-cast sample. Annealing of these samples further decreased their hardness, however, by only 3 to 7% compared to respective as-cast states of each alloy.

## 4. Discussion

Sn was chosen to supplement the composition of ZnAlMg based alloys. Similar Zn-based alloy compositions have not been reported in the literature so far, probably due to the concerns related to the corrosion properties of such alloys. These properties will be investigated in adjacent research.

Different aspects of the alloy’s microstructure were investigated. The main phases identified were in the general agreement with the published data as follows: η(Zn), α(Al), Mg_2_Zn_11_, MgZn_2_ [7,8,9,10,11,12,13,14,15,16,17,18,20,21,22,23,24] and Mg_2_Sn [26,27,32,33].

DSC curves can be clearly described only by investigating the microstructure of DSC samples after cooling in the DSC equipment (Figure 2, Figure 3 and Figure 4). Copper mold as-cast microstructure is an order of magnitude finer and hence less likely to be clearly identified. A clear comparison can be for example given by comparing the size Mg_2_Sn/η(Zn) eutectic particles in Figure 4 (DSC sample of MZ + 3.0Sn) and Figure 9e (as-cast sample of MZ + 3.0Sn). Furthermore, the kinetic of solidification may also affect the order in which the phase or phase mixtures are formed. DSC curves show that the ternary η(Zn)/α(Al)/Mg_x_Zn_y_ eutectics of MZ + 0.0Sn alloy are replaced by quaternary η(Zn)/α(Al)/Mg_x_Zn_y_/Mg_2_Sn eutectics by adding 1 wt.% of Sn. By gradually adding Sn from 0.0 to 1.0 wt.%, the peak of the quaternary eutectic areas is formed at ~335 °C, while the peak of the ternary eutectic areas, found at temperatures in the range from 347.5 to 339.1 °C, is being gradually suppressed. Further addition of Sn (2.0 and 3.0 wt.%) causes the ternary reaction peak to shift towards even lower temperatures and being completely overlapped by the quaternary reaction peak. This is in line with available assessment of liquidus projection for the Zn-Mg-Sn ternary system [32,45]. These systems also predict a decrease in liquidus temperature for less complex eutectics when Sn concentration is approaching a more complex eutectic point near the Zn-rich corner of this system.

Peaks at 357–346.3 °C represent the binary eutectics. While DSC curves suggest only a gradual peak shift of binary eutectic reaction, the microstructure investigation showed that Mg_2_Zn_11_/η(Zn) binary eutectics is being replaced by Mg_2_Sn/η(Zn) in case of the MZ + 3.0Sn alloy. Based on available ternary Zn-Mg-Sn assessments [32,45], it is hypothesized, that Sn supports the preferential formation of Mg_2_Sn/η(Zn) binary eutectic instead of Mg_2_Zn_11_/η(Zn) eutectic mixture. This is observed in the currently investigated system as well, despite the presence of Al as an additional alloying element. It is also worth mentioning that the temperature difference between the binary and ternary eutectic points calculated [32,45] is only 1 °C. This may cause difficulties to reveal the real order of solidification reactions since even a slight local chemical difference or temperature heterogeneity may cause local fluctuation and a competitive formation of Mg_2_Sn/η(Zn) and Mg_2_Zn_11_/η(Zn) binary eutectic areas as indicated for the MZ + 2.0Sn alloy.

Heating curves shown in Figure 1 depicted also the peaks corresponding to the α(Al) + η(Zn) → γ(Al) eutectoid transformation at ~285 °C. The corresponding reaction cannot be observed during a cooling DSC run since this eutectoid transformation is rather sluggish; therefore, it is without a detectable heat release. The microstructure investigation shows that the γ(Al) → α(Al) + η(Zn) reaction clearly occurs; however, probably over a much broader temperature range compared to the heating curves. This reaction might be finished even at an ambient temperature [46].

Adding Sn reduced the volume content of Mg_x_Zn_y_ intermetallic phases, and these were replaced by Mg_2_Sn particles. This behavior is in line with the literature findings on similar systems [27,32]. The addition of Sn mainly affects the volume content of Mg_2_Zn_11_ (Figure 7).

For similar Zn-Al-Mg alloys, the sources report the same two Mg_x_Zn_y_ phases to be present [21,22,25,47]. Vlot et al. [47] identified only MgZn_2_ in similar alloys, while other literature sources confirmed the presence of both phases mentioned [21,22,25]. In our samples, Mg_2_Zn_11_ is the primary phase; however, MgZn_2_ was also clearly identified by both XRD and even SEM/EDX. At 3.5 vol.%, its content was highest in the as-cast MZ + 0.0Sn sample. As MgZn_2_ is a non-equilibrium phase in the current system, its content is significantly reduced by annealing the basic MZ + 0.0Sn alloy at 310 °C for 1 h.

Mg_2_Zn_11_ and MgZn_2_ are competing phases and their final ratio is complex to predict and control even in a simple Mg-Zn alloy as reported by several sources [23,24]. All in all, their presence will depend on several factors like exact alloy composition or cooling rate [48].

The amount and distribution of intermetallic particles appears to have a direct influence on the microhardness of the studied alloys. Overall volume content of intermetallic phases is decreasing with the increasing wt.% of Sn as measured by XRD (Figure 7 and Figure 8) as well as the SEM image analysis (Figure 10). This is reflected in the decreasing alloy hardness summarized in Figure 11.

Formation of Mg_2_Sn particles and the gradual increase of their vol.% is reported to cause an increase in hardness for specific Mg-based alloys with similar Sn content [29,30,31]. The same mechanism does not apply to our Zn-based alloys, as in our samples, the overall vol.% of intermetallics is decreasing.

The hardness of as-cast samples is marginally higher compared to annealed samples (Figure 11). This is most probably caused mainly by the change in shape and distribution of the intermetallic particles. This can be observed when comparing the images of as-cast vs annealed conditions in Figure 9. The annealing allows η(Zn) to diffuse from eutectics in the interdendritic areas towards the primary η(Zn) dendrites, hence changing the eutectic nature of the interdendritic areas. For the MZ + 0.0Sn and MZ + 0.5Sn alloys, the original dendritic character of the microstructure can still be recognized even after annealing Figure 9a vs. Figure 9b,c vs. Figure 9d. For higher Sn content, this is not possible. The addition of 1 to 3 wt.% of Sn into this alloy system, enabled a more effective spheroidization and agglomeration of individual phase particles. Hence, the annealing had a more apparent influence on the microstructure of these alloys (MZ + 1.0Sn, MZ + 2.0Sn, MZ + 3.0Sn).

For as-cast and annealed samples a different ratio of microstructural components was established for the same alloys by SEM. Nevertheless, both as-cast and annealed samples show the same trend compared to the XRD quantitative analysis. Additionally, for the annealed samples, the SEM image analysis and XRD analysis are in even better agreement.

## 5. Conclusions

Melting of MZ + 0.0Sn starts at 344 °C, while with the addition of 0.5–3.0 wt.% of Sn, melting starts already at 334 °C. Melting is finished at 382 °C for the MZ + 0.0Sn and this temperature is being continuously decreased to 376 °C by the addition of up to 3 wt.% of Sn.

Main phases identified in the MZ + 0.0Sn alloy were η(Zn) and α(Al) solid solutions as well as Mg_2_Zn_11_ and MgZn_2_ intermetallic phases. Addition of Sn enabled the formation of Mg_2_Sn intermetallic phase at the expense of Mg_x_Zn_y_ phases, while mainly affecting the vol.% of Mg_2_Zn_11_.

The microstructure is dendritic for all as-cast alloys. The interdendritic areas are formed by the binary, ternary and quaternary eutectics specific for each alloy. Alloying with Sn causes the following changes of microstructural components: ternary eutectics consisting of η(Zn), α(Al) and Mg_x_Zn_y_ phases are gradually replaced by quaternary η(Zn), α(Al), Mg_x_Zn_y_ and Mg_2_Sn eutectics. Binary η(Zn) + Mg_x_Zn_y_ eutectics are gradually replaced by binary η(Zn) + Mg_2_Sn eutectics.

For the MZ + 0.0Sn and MZ + 0.5Sn alloys, the original dendritic character of the microstructure can still be recognized even after annealing. At the same time, the individual phases from the eutectics are connected to discrete particles, and thus the original eutectics are not recognizable anymore. Introducing 1 to 3 wt.% of Sn into this alloy system enabled a more effective spheroidization and agglomeration of individual phase particles significantly changing even the shape of the primary η(Zn) dendrites.

Annealing causes slight changes in the phase composition. For MZ + 0.0Sn mainly MgZn_2_ is transformed to Mg_2_Zn_11_. For the alloys with Sn, the volume content of Mg_2_Zn_11_ is partially increased mainly at the expense of Mg_2_Sn.

The microhardness is decreasing with the increasing of Sn content. The annealing changes the microhardness only slightly.

Based on microstructure observation, these alloys are overall suitable for coatings exposed to extended high temperature exposure. As coatings of steel substrates, their corrosion properties will be at least maintained as reported in part two of this research: The effect of Sn addition on Zn-Al-Mg alloy-Part II.

## Figures and Tables

**Figure 1 materials-14-05404-f001:**
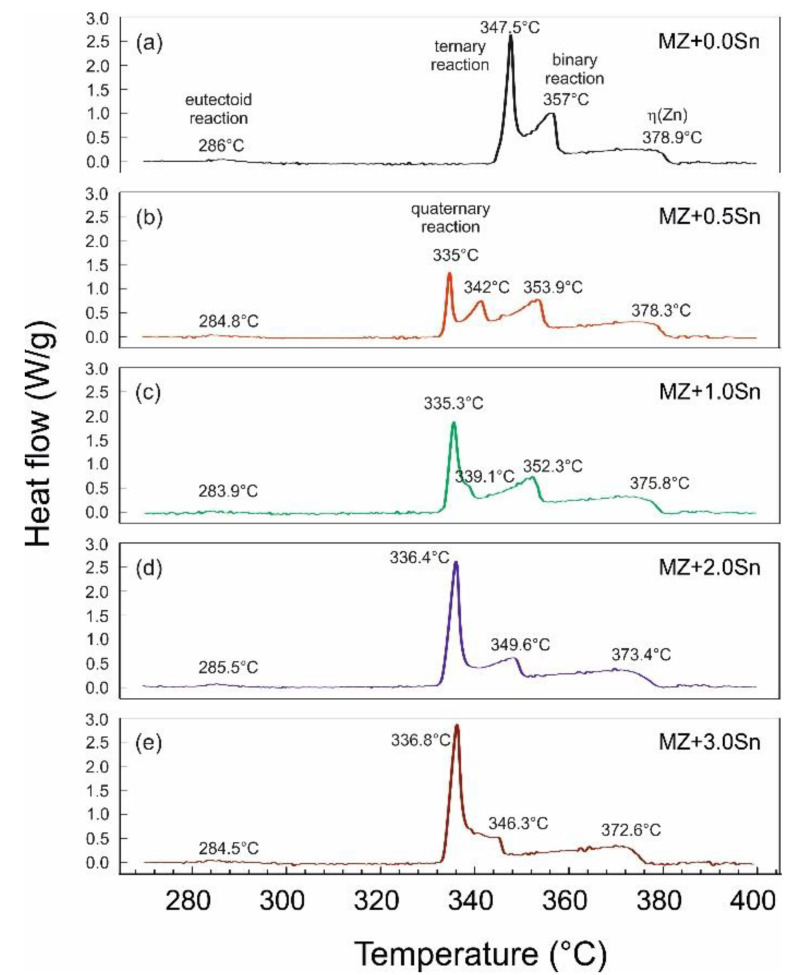
Comparison of the DSC curves for all investigated alloys.

**Figure 2 materials-14-05404-f002:**
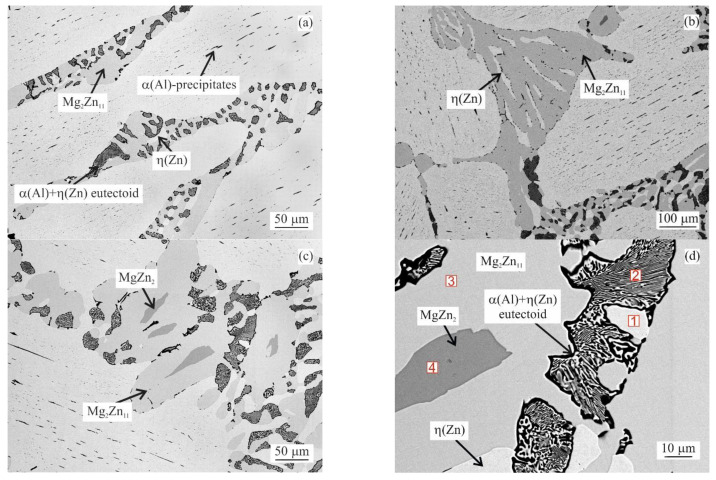
Selected features of MZ + 0.0Sn alloy microstructure after DSC measurement: (**a**) example of ternary eutectic; (**b**) example of binary eutectic; (**c**) example of Mg_2_Zn_11_ and MgZn_2_ mixture; (**d**) area with EDS measurement points 1–4 listed in Table 2.

**Figure 3 materials-14-05404-f003:**
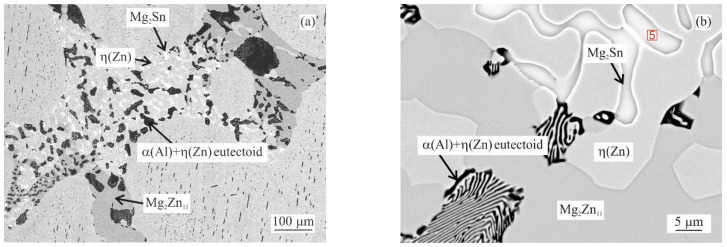
Quaternary eutectic area in MZ + 0.5Sn alloy microstructure after DSC measurement: (**a**) example of quaternary eutectic; (**b**) closer detail of such area including EDS measurement point 5 listed in Table 2.

**Figure 4 materials-14-05404-f004:**
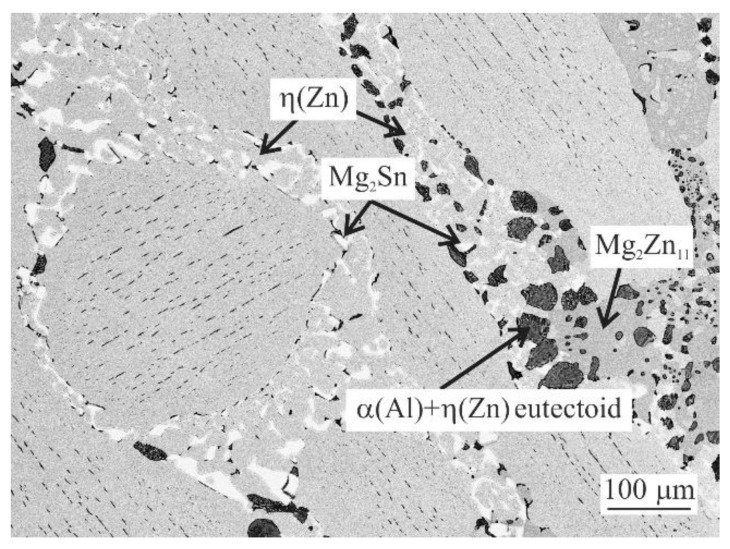
Binary and quaternary eutectic areas in MZ + 2.0Sn alloy microstructure after DSC measurement.

**Figure 5 materials-14-05404-f005:**
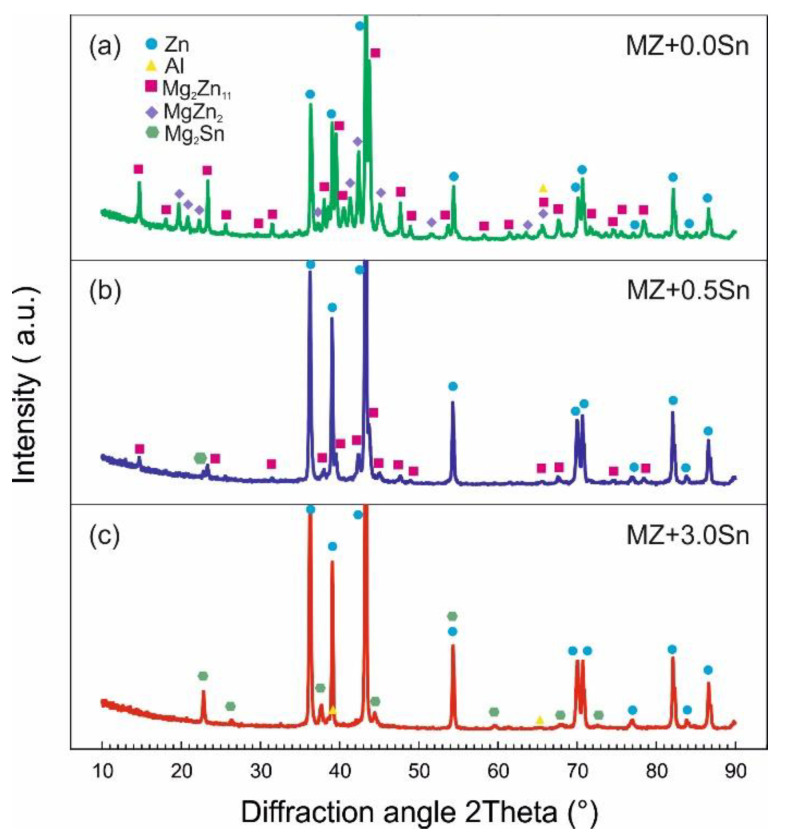
XRD diffraction patterns recorded on powder samples of selected alloys in as-cast state (**a**) MZ + 0.0Sn, (**b**) MZ + 0.5Sn, (**c**) MZ + 3.0Sn.

**Figure 6 materials-14-05404-f006:**
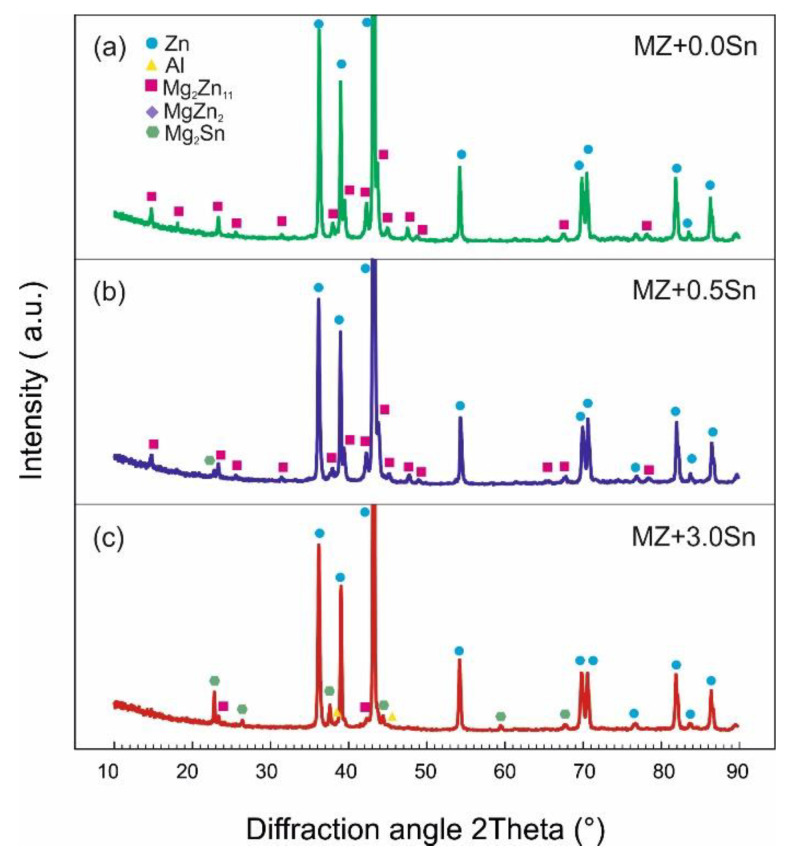
XRD diffraction patterns recorded on powder samples of selected alloys in annealed state (**a**) MZ + 0.0Sn, (**b**) MZ + 0.5Sn, (**c**) MZ + 3.0Sn.

**Figure 7 materials-14-05404-f007:**
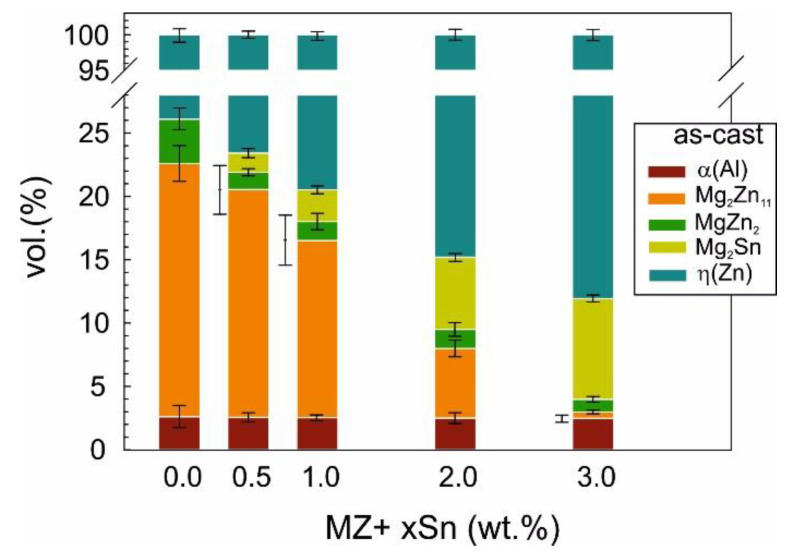
Phase composition in vol.% in metallic bulk samples as measured on metallic filings from as-cast samples.

**Figure 8 materials-14-05404-f008:**
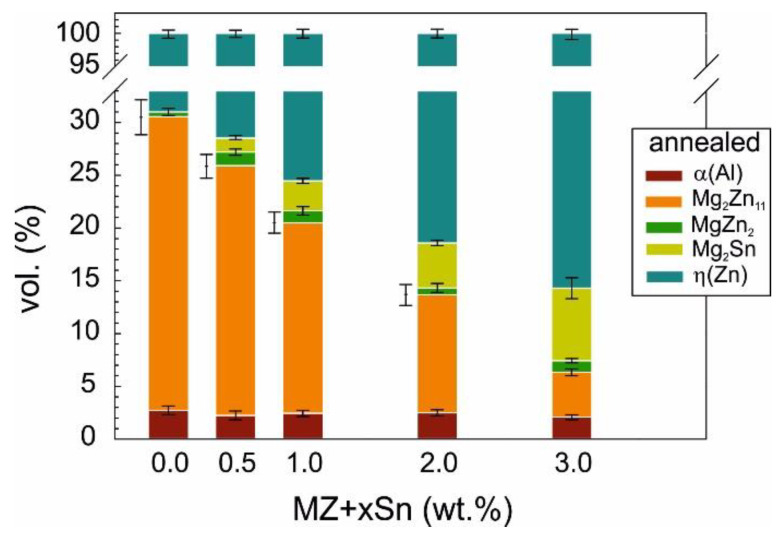
Phase fractions in vol.% in metallic bulk samples as measured on metallic filings of annealed samples.

**Figure 9 materials-14-05404-f009:**
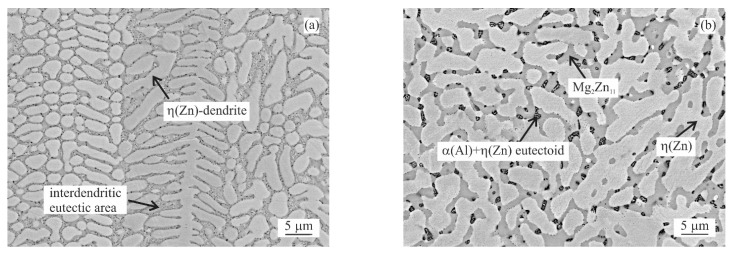
Microstructure images for selected samples, longitudinal section, near sample surface: (**a**) MZ + 0.0Sn as-cast, (**b**) MZ + 0.0Sn annealed, (**c**) MZ + 0.5Sn as-cast, (**d**) MZ + 0.5Sn annealed, (**e**) MZ + 3.0Sn as-cast, (**f**) MZ + 3.0Sn annealed.

**Figure 10 materials-14-05404-f010:**
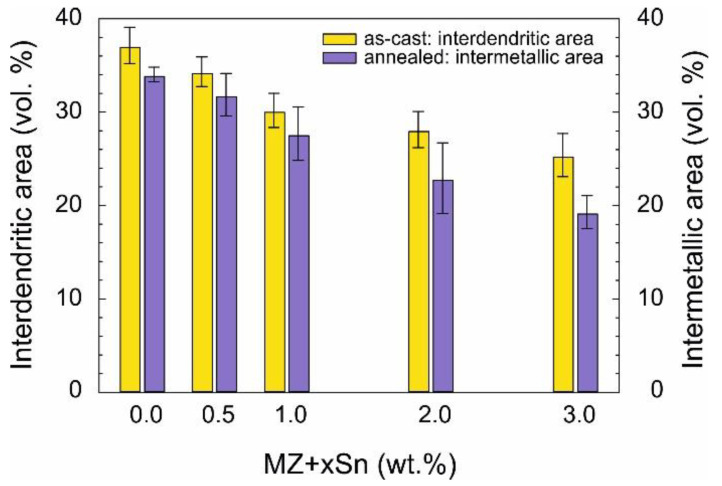
Vol.% of microstructural components as determined from SEM image analysis.

**Figure 11 materials-14-05404-f011:**
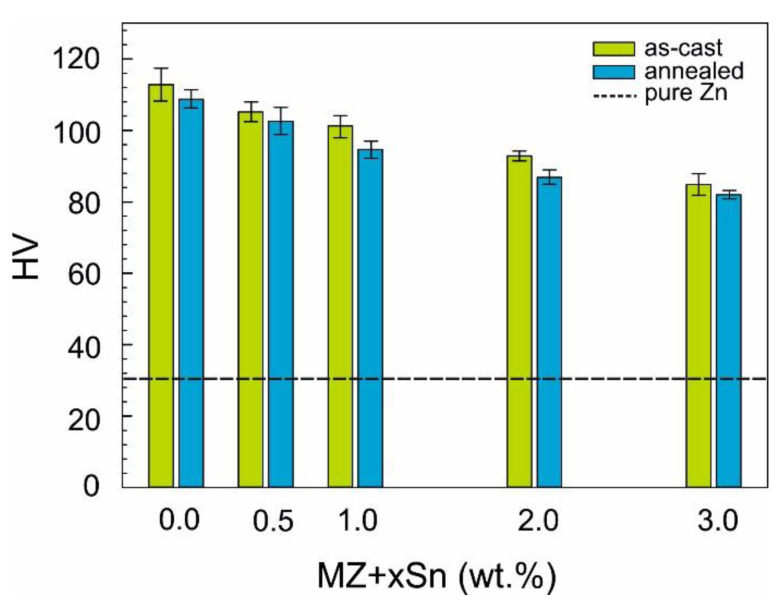
HV1 microhardness of the investigated samples.

**Table 1 materials-14-05404-t001:** Chemical composition of the studied alloys (wt.%).

Alloy	Al	Mg	Sn	Zn
MZ + 0.0Sn	1.56 ± 0.07	1.40 ± 0.01	0.07 ± 0.02	bal.
MZ + 0.5Sn	1.64 ± 0.02	1.41 ± 0.01	0.52 ± 0.01	bal.
MZ + 1.0Sn	1.62 ± 0.03	1.45 ± 0.02	1.06 ± 0.02	bal.
MZ + 2.0Sn	1.57 ± 0.01	1.44 ± 0.01	1.95 ± 0.01	bal.
MZ + 3.0Sn	1.57 ± 0.12	1.43 ± 0.05	2.69 ± 0.06	bal.

**Table 2 materials-14-05404-t002:** EDS chemical composition of selected sites (at.%).

	Site No.				
Chemical Element (at.%)	1	2	3	4	5
Zn	99.20	54.50	83.70	66.30	66.10
Al	0.80	45.50	-	-	-
Mg	-	-	16.30	33.70	-
Sn	-	-	-	-	33.90
Phase/Region	η(Zn)	α(Al) + η(Zn) eutectoid	Mg_2_Zn_11_	MgZn_2_	Mg_2_Sn

**Table 3 materials-14-05404-t003:** Phases identified during XRD analysis.

Phase Chemical Formula	Reference Code–ICSD Database FIZ Karlsruhe	Crystal System	Space Group	Space Group Number
η(Zn) = Zn + 2 at.% Al	98-024-7160 modified according [42]	Hexagonal	$P6_{3}/mmc$	194
α(Al) = Al + 14 at.% Zn	98-060-6001 modified according [42]	Cubic	$Fm\bar{3}m$	225
MgZn_2_	98-010-4897	Hexagonal	$P6_{3}/mmc$	194
Mg_2_Zn_11_	98-010-4898	Cubic	$Pm\bar{3}$	200
Mg_2_Sn	98-064-2855	Cubic	$Fm\bar{3}$	225

## Data Availability

Data sharing is not applicable.

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
