# Peer review of "The Effect of Sn Addition on Zn-Al-Mg Alloy; Part I: Microstructure and Phase Composition"

_materials, 2021, doi:10.3390/ma14185404_

Round 1

Reviewer 1 Report

Overall, this is a well written and constructed piece of work covering a good range of analytical techniques for incremental variations on the Zn-Al-Mg alloy, particularly with the focus on a group 14 elemental addition. Generally, a good bit of work but I think there are a few minor corrections that should be included to improve the quality of this work.

Line 91/92: Can you explain why it was unreliable? And why you used filings instead of powdering/ball milling the sample to get a more representative sample that is more reliable quantitatively, as you are then completing Rietveld refinement afterwards?

94: This is one of the more in-depth XRD sections I've seen in methods so well done on that. I'd just say that it's not referred to as an XRD lamp but as a source. In this case a Cu radiation source run at 40kV and 40mA. Given you are doing Rietveld refinement I’d also include information on any slits/sollers used in the setup and step size/time per step just to flesh it out fully.

101/102: Given the details you’ve gone into with the XRD I’d suggest adding some more information to the SEM section, for instance: any settings of kV and mA used for the imaging and what the source is/what the samples were mounted in.

121/Figure 1: I will be worth labelling the peak relationships directly on the figure. just to tie it into the text better. Hopefully the resolution is due to the format I have been given, but if not, please ensure the images are high resolution so they look less pixelated and distorted.

133/Figure 2: This figure could really do with rearranging, so it looks neater. Not sure if it's just due to the format I'm seeing it in but the 3 images just looks a little strange, Adding a 4th in would be helpful for symmetry. In addition, the 7800 used is capable of going to much higher zoom levels and a higher zoom onto the α+ɳ eutectoid sections or the ɳ (Zn) that is just about visible in b would be really helpful for the understanding

140-145/Figure 3 and 4: Again this would benefit from a further zoomed in image on the Mg2Sn structures formed here. Looks to be some form of cracking happening around the phase in figure 3. Both figures exhibit the darker "dashes" within the dendrites, what these are and why they occur need discussing, Are they due to polishing or some form of Al rich phase? This kind of defect my affect the mechanical and corrosion properties so it’d be useful to know what they are.

On all SEM images: It is unfortunate there is no EDS analysis undertaken on these images. It would be beneficial to the story to show some elemental maps. It would help show the smaller features briefly pointed out in a better light.

177/Figure 5 + 6: There are a large number of phases here so I'm hoping it's just this document compressing the images, but if not please ensure these are higher resolution so a reader can fully zoom in to see what is going on here. But they are generally neat XRD graphs.

The data in figures 5 (b) and 6 (b) are so close to each other in comparison to their counterparts, a bit of clarity as to why this is would help.

183: this might just be formatting for the reviewer copy but this single sentence below the XRD graphs just looks scruffy

216: If you are stating Rietveld refinement for this kind of volume fraction more information on the method needs to be given. For starters what peak shape function are you using (Pseudo-Voigt etc), how reliable is the data using R factors and GOF?

It is also useful to show some of the calculated models to show how they actually fit and what the difference is (this can be supplementary if needed)

248-250: I think you said the data was taken from 15 images on each sample? How consistent was this and could you give a +/- to each value to better show the variability?

What are the error bars on the charts done to?

258-261: Similarly with the microhardness values, how many points were taken to get each bit of data and what are the error bars showing here (full data range? Or SD?)

265-268: I would disagree that this alloy has not been reported in literature thus far. You've mentioned at least one in your references (27) Whilst they are not typically in the exact same composition as you are looking at say it's definitively worth acknowledging that other works on similar ZnAlMg+Sn alloys has been done already.

Further references examples include: "Evolution of tension and compression asymmetry of extruded Mg-Al-Sn-Zn alloy with respect to forming temperatures" Sang-Ik Lee and "Microstructure and mechanical property of high strength Mg–Sn–Zn–Al alloys" Yun Li

Whilst I am aware these are not looking at the exact composition you are investigating it's worth mentioning them.

Work has also been done on similar alloys with Mg2X forming elements like Ge and Si. Worth mentioning these as well as they inevitably exhibit similar structural changes as a result of the Mg reacting with the group 14 elements (Ge+Si). Might be worth just touching on these to add a greater context to the work and some comparison of the impact of these group 14 elements. For instance: "Increased Corrosion Resistance of Zinc Magnesium Aluminum Galvanised Coating through Germanium Additions" Shahin Mehraban. I don’t think there has been any work on the Si additions published but I’ve only had a short literature scan.

317-320: Is there any evidence of localised cracking within these structures during the microhardness tests? MgZn2 is a characteristic hard and brittle phase, and is the reduction in this phase going to make the material more formable as it does with other Mg scavenging elements added to this kind of alloy?

Conclusion: Overall you’ve shown and said what this alloy addition has done but it would be worth adding some context as to what this might mean for the alloy in its applications. I’m presuming you are planning on doing a part 2 covering the corrosion aspects but it’ll be worth mentioning it here if just to add context to the work.

Author Response

Dear Reviewer,

Thank you for the highly inspiring comments.

Please, see your review attached with our responses to each of your comments.

We have implemented them to the greatest possible extend.

Best regards,

Martin Kusy

Reviewer 2 Report

Dear Authors,

The paper sent to review will be interested for Materials journal readers. The reviewed paper should be considered as being at a very good level. Especially for methodology and range and quality of elaborated tests. Title and keywords are appropriate and adequate to paper content. The results are presented comprehensively. The literature review is up-to-date.  The goals of the paper are sufficiently explained. The conclusions are adequate.

However, the authors did not avoid some flaws which should be corrected and/or supplemented:

- in Abstract add information what is the novelty of this study, please (this information is in the text - first paragraph in section 4),

- in Introduction, at the end, it should be highlighted what is the aim of this study,

- p.1 l.33 - too many citations of one author - i.e. Nishimura,

- p.2 l.82 - add standard, please - e.g. EN ISO 6507...,

- p.2 l.85 - is room temperature, better would be ambient,

- p.6 l.146, 150, 155 - is (Figure 1c))... should be (Figure 1c)...

- p.11 l.271 - too many citations - split if possible, please,

- p.15 l.478 (reference 41) - is zn-al  should be Zn-Al.

Best regards

Author Response

Dear Reviewer,

Thank you for the helpful comments.

Please, see your review attached with our responses to each of your comments.

We have implemented them to the greatest possible extend.

Best regards,

Martin Kusy
